# The Use of Sleds as a Unique Training Technique for Anaerobic Performance Development among Young Basketball Players

Roni Gottlieb [1,*], Aviad Levi [2], Asaf Shalom [3], Julio Calleja Gonzalez [4] and Yoav Meckel [1]

1  School of Human Movement and Sport Sciences, The Levinsky-Wingate Academic College (Wingate Campus), Netanya 4290200, Israel; meckel@l-w.ac.il
2  Bnei Herzliya Basketball Club, 1st Division Israeli League, Herzliya 46497, Israel; aviadlevi86@gmail.com
3  Department of Physical Education, Tel Hai Academic College, Qiryat Shemona 1220800, Israel; asaf.fitness@gmail.com
4  Department of Physical Education and Sports, Faculty of Education and Sport, University of the Basque Country (UPV/EHU), 01006 Vitoria-Gasteiz, Spain; julio.calleja.gonzalez@gmail.com
*  Correspondence: ronigot23@gmail.com; Tel.: +972-54-482-2425

**Abstract:** (1) Playing basketball entails intense anaerobic activities, including short sprints, sudden stops, rapid direction changes, and jumps. Common training methods for enhancing players' anaerobic fitness comprise sprint training, jump exercises, and interval training. The aim of this research was to compare the impact of three sprint-training methods on anaerobic capacity. (2) The study included 25 male basketball players, aged 16–18 years, from the National Youth League in Israel. The participants were divided into three groups: sled-pushing, sled-dragging (intervention), and no sled (control) sprint training. Twice-weekly sessions were conducted over a five-week period, in addition to their regular basketball training. Pre- and post-intervention tests included vertical and horizontal jumps, 5 and 20 m sprints, and agility tests. (3) The study revealed significant improvements in the horizontal jump in the sled-pushing group (≈3%) and a near-significant improvement in the sled-dragging group (≈2.9%). Less improvement was seen in the control group (≈1.8%). No improvements were seen in the vertical jump performance in any groups, despite an increase in the sled-pushing group (≈5.5%) and in the sled-dragging group (≈3%) compared to that in the control group (≈1.2%). Finally, no improvements were seen in sprint speed and agility in any group. (4) Despite the modest improvements observed in most tests across the three groups, the consistent and significant enhancement seen in the sled-pushing and sled-dragging groups compared to that in the control group suggests potential benefits for sled assistance in anaerobic training in young basketball players.

**Keywords:** explosive; basketball; sled; training methods; anaerobic development



## 1. Introduction

Anaerobic fitness plays a fundamental role in the performing of short and intense activities, which are common in sports such as sprinting, gymnastics, and various ball games [1,2]. The anaerobic system is based on metabolic processes that occur in active muscle cells and rely on local carbohydrate stores (without the use of oxygen) [1]. Moreover, the anaerobic system is divided into two pathways: (1) the ATP-CP (a lactic) pathway, which primarily provides energy for performing short and intense activities (0–10 s) such as sprints, jumps, and throws [1,3]. This pathway entails the breakdown of both ATP molecules and stored energy molecules known as creatine phosphate (CP), which is found in muscles in small quantities; and (2) the glycolytic pathway, which supplies the energy needed for performing intense or prolonged activities [2,4]. This pathway is dominant in more intensive efforts that last between 10 s and 3 min. Lactic acid—the byproduct of glycolysis—increases in the muscles during intense physical activities. Moreover, the

glycolytic system provides energy for activities such as interval training, yet it does not directly cause the fatigue that is experienced during the activity [5].

### 1.1. The Impotance of Anaerobic Fitness Components in Basketball

The game of basketball is characterized by short and intense bursts of activity combined with periods of low-intensity efforts or complete rest. Many basketball-related movements rely on anaerobic capabilities, such as short sprints, accelerations, decelerations, changes in direction, and horizontal and vertical jumps [6–8].

Using modern technology, the actions of players can be tracked and monitored during the game [9,10]. In a study on a basketball team in the Spanish premier league, researchers found that during one game, basketball players executed an average of 15 high-intensity actions [11]. These included about 2–3 sprints (3.5 m/s), 2 decelerations (3.5 m/s), 1 jump (>40 cm), and 8–10 changes of direction. Moreover, guards tended to perform more changes of direction than forwards or centers.

In basketball, high running speeds are mainly required during the transition from offense to defense and vice versa, where players perform about 50–80 short-distance sprints during a game. In a study on Australian basketball players, researchers found that the players spent about 70–80% of the game running at low speeds (0–6 km/h), 10–15% at moderate speeds (6–19 km/h), and 5–10% at high running speeds that were greater than 19 km/h [12].

When examining metabolic changes during a game, elite basketball players were found to accumulate 4–6 mill of lactate per 1 L of blood. Such findings suggest that the activation of the glycolytic–anaerobic energy system while playing basketball is insignificant, and the accumulation of lactate in muscles does not significantly limit the players' intense activities [13]. However, in basketball players, the contribution of the aerobic system is to the facilitating of quick actions during the game [2,14].

The aerobic system also plays a role in recovery between intense efforts [2,15,16]. In a study on youth basketball players, a correlation was seen between the predicted VO2max and the players' performance in repeated sprints, with better aerobic capacity being linked to faster recovery [17]. Professional basketball players typically have a VO2max of 50–60 mL/kg/min [15,16], with shorter players often having higher aerobic capacities. During full games, players maintain an average of around 85% of their maximum heart rate [18].

Muscle strength significantly affects athletes' acceleration capabilities. In a study on professional basketball players, improved strength (seen through the squat exercise test) resulted in improved 5 and 10 m sprint performance [19]. Moreover, combined strength training, which focuses on eccentric work among basketball players and others, was found to be crucial for improving explosiveness, agility, and direction change abilities. Due to their high relevance to the game of basketball, such training should be integrated into the training regimen of basketball players in general [20]. Yet trainers should note that prolonged submaximal strength training sessions could lead to increased muscle mass and decreased explosive power [1,20–22]—which could hinder, rather than improve, acceleration, agility, and maximum speed in athletes.

Finally, the significance of explosive strength in basketball was highlighted in a review article, whereby a strong relationship was seen between explosive strength and most actions performed by basketball players on the court, such as sprints, direction changes, stops, and jumps [1–4]. Players with a stronger lower body were found to possess better explosive power and, therefore, minimize ground-contact time—while still producing maximal eccentric force in subsequent jumps, allowing them to secure rebounds faster [23]. Consequently, it is crucial that attention is paid to players' explosive strength during training, as this could contribute to the efficient performance of players on the court [1,3].

### 1.2. Developing Anaerobic Fitness Capabilities

Due to the demanding nature of anaerobic activity and the need for relative recovery, training sessions should be based on the pyramid principle, entailing alternating activity–rest cycles [14,17]. Additionally, the training approach for developing anaerobic fitness should suit the specific discipline as well as the athlete's expertise [2,17].

As anaerobic workouts are often of high intensity, achieved through the application of substantial loads on the muscular and skeletal systems, a focus should be placed on muscle strength development [24]—i.e., resistance training. Although such training can be achieved through a range of methods, weight training is often the preferred method. Such exercises typically involve dynamic muscle action, performed against resistance in concentric and eccentric phases. During these actions, the weights, repetitions, and speeds should be tailored to the athlete's specialization and training phase [3,25,26].

Additionally, anaerobic training places an emphasis on explosive power, enhanced through exercises with submaximal resistance levels or body weight. These workouts entail swift movements, involving the body or external objects such as weights, sleds, and power balls, for impacting both the muscular system and the nervous one [3,27,28].

Specifically in the sport of basketball, improvements in players' explosive power can be achieved through manipulations of high-speed training within each phase of resistance—such as through interventions that emphasize high-speed concentric phases [23]. However, trainers and athletes should be cautious, as exercises emphasizing eccentric phases may lead to the delayed onset of muscle soreness, potentially affecting performance in the days following the workout [29].

### 1.3. Explosive Power Development Training Methods

In light of the importance of developing explosive power in athletes in general and in basketball players in particular, a range of methods have been developed and applied over the years [1–3]. For example, *explosive jumps* can be developed through plyometric training, based on a rapid and strong contraction of the muscle immediately following a quick and sudden stretch. As this stretch precedes the contraction action, this triggers the stretch-shortening reflex in the muscle [30–32]. Activation of this reflex assists in producing a powerful contraction force—one that is greater than the force generated in a regular contraction state without a prior rapid muscle stretch [3,32]. Improving this functional component aids in a stronger and faster push against the ground—in various jump performances or during rapid ground-contact in short sprints. Common plyometric exercises involve performing jumps over hurdles, boxes, or jumping and landing from different heights—in line with the specific sport and the athlete's capabilities. A typical training session includes sets of leaps, totaling around 50–150 leaps, with rest intervals of about 2–4 min between sets [24,31].

In a study on rugby players, the impact of specific plyometric training (via bounding exercises of both horizontal and vertical jumps) was compared to that of stand-alone sprint training and a control group over an eight-week period. The research revealed that the increase in the participants' maximum 40 m sprint speed in the two former groups was comparable to that in the control group. Moreover, the most significant impact of the plyometric training was seen in the initial acceleration phase—i.e., the initial 10 m of the run, where the plyometric training group notably exhibited the greatest speed improvements compared to the other two groups [33]. In an additional study on basketball players, a 10% body weight load was added to the intervention group participants, compared to that of the control group members, who performed the same plyometric exercises yet only with body weight. After the 10-week intervention program, significant improvements were seen in vertical and horizontal leaping in both training groups; however, the intervention group exhibited significantly greater improvement compared to the control group [30].

In a study on female high-school basketball players, two plyometric training programs were compared: forward jump exercises versus lateral jump ones. The former exhibited significantly improved forward jumps, while the latter presented greater improvements in

both lateral jumps and in directional changes that were also performed on the lateral plane. Hence, in multidirectional sports such as basketball, the integration of specific plyometric exercises into various planes of motion is crucial [34].

It is important to differentiate between leaps and jumps. While leaps occur as a cyclical movement—vertically or horizontally or forward or laterally, jumps are performed as a cyclical running movement and only along the horizontal plane. When jumping, the leg muscles exert significant force against the ground—as a means for rapidly propelling the body forward. In the initial acceleration phase (0–20 m), it is the muscle-contraction force that primarily affects the movement; in the latter acceleration phase (20–40 m), the speed of contraction becomes the dominant factor. As such, jumps of up to 30–40 m (lasting about 4–5 s) can be an effective training tool for impacting both the muscular system and the nervous one. In plyometric power training sessions with jumps, 8–12 jumps are commonly performed, with 2–3 min rest intervals between each jump [3,31,35]. During jump training, additional resistance can be added, using aids such as resistance bands, weighted vests, and sleds [24,36,37]. However, increasing loads through resistance aids may compromise movement techniques, embedding incorrect movement patterns in athletes.

In a study on rugby players, the effect of specific plyometric training (a combination of horizontal and vertical jumps) compared to only sprint training and a control group that performed only rugby training for 8 weeks was tested on different anaerobic performances. It was found that the running-specific plyometric training group (sheep steps) had improved maximum speed in a 40 m sprint to the same extent as the sprint training group, and both significantly more than the control group. Furthermore, it seemed that the most significant effect of the plyometric training was evident in the initial acceleration phase—the first 10 m of the run, where the plyometric training group significantly improved speed more than the sprint training group and the control group [33]. In another study involving soccer players, four training methods for developing explosive strength were evaluated: weight training, plyometric exercises, resistance jumps (using a sled), and no resistance. The research found an 8–10% improvement across all four training methods in various anaerobic performances, such as jumps—with and without added weight. However, only weight training was found to notably improve lower limb strength and stride length in jumps. As such, there seem to be diverse methods for enhancing explosive strength and acceleration, as there is a need for strength training alongside other forms of exercise [38,39].

Studies have examined the impact of sled training on vertical and horizontal jumps, short sprints, and various agility tasks [28,36,40,41], yet few have compared between sled pushing, sled pulling, and alternative methods such as body weight resistance. The aim of this study was to evaluate the impact of three sprint-run training methods on the anaerobic abilities (explosive power, maximum speed, and agility) of young male basketball players. The methods included sled-pushing (at 40% body mass), sled-pulling (at 40% body mass) and non-sled training. Based on the literature review presented above, the following two research hypotheses were defined:

1.  Significant improvements will be seen in the participants' anaerobic abilities following the sled training; similar improvements will be seen in the anaerobic capacity of the two intervention groups compared to that in the control group.
2.  The impact of sled pushing and pulling at 40% body mass will have a higher effect compared to repetitions without sled use on anaerobic components such as explosive power, maximum speed, and agility among young basketball players. There will not be a difference in the impact between sled pushing and sled pulling workouts among the players.

## 2. Materials and Methods

A total of 25 male basketball players aged 16–18 years participated in the study. They were members of the 2 youth teams of the Rishon LeZion Basketball Association in Israel and had 8–9 years of basketball experience. For the past four years, in addition to their

basketball training, they had also participated in two weekly 60 min strength-training sessions with weights in the gym, and in one weekly 45 min athletic training session. All players followed the same training regime, and none had no previous experience in sled training. The inclusion criteria were as follows: (1) no physical injury over the two months leading up to or during the intervention; (2) they had participated in at least 80% of the training sessions in the two months leading up to or during the intervention; and (3) they had at least 5 years of basketball experience.

### 2.1. Procedure

Prior to and during the intervention, the participants trained four times a week. The trainings were conducted at the Rishon Lezion Sport Centre (Rishon Lezion, Israel) as part of the preparatory program for the upcoming season. The research was conducted in the following three stages:

*Stage 1. Pre-intervention tests.* In the first week of the preseason, the participants underwent anaerobic fitness tests in the facility where they regularly trained. The tests were conducted on parquet flooring after a two-day rest period and were preceded by a fifteen-minute warm-up. The tests examined explosive strength in horizontal and vertical movements, maximum speed, and agility. Each test was performed twice, and the best result was recorded. The tests included protocols for explosive strength, maximum speed, and agility, with resting intervals for preventing fatigue. The participants were then divided into three groups, while ensuring a similar heterogeneity of explosive strength between the groups.

*Stage 2. Intervention.* The three groups underwent twice-weekly training sessions (on Sundays and Wednesdays) over a five-week period. These entailed high-intensity interval training of short-distance sprints, as follows. **Group A**: dragging a sled at a weight equal to 40% of their body weight; **Group B**: pushing a sled at a weight equal to 40% of their body weight; and **Group C**: regular, non-sled running, with only body weight resistance. The training sessions were conducted on an asphalt court in the late afternoon hours, with up to 70% humidity, at least 3 h post-meal, and with players wearing their standard basketball gear. Each session began with the same active warm-up, comprised of aerobic activity, dynamic stretching, specific motor preparation, and two 30 m accelerating sprints. The training also entailed short-distance sprints at maximum speed with high-intensity bursts from a static position. All groups underwent the same training regimen. Maximum effort and straight-line running were emphasized, with the coaching staff and researchers providing continuous instruction and motivation throughout the training. Dates for conducting the pre- and post-intervention tests were set in advance, so as not to interfere with training and competitions. Table 1 presents the performance variables of the repetition training for all three groups.

**Table 1.** Performance variables of the intervention training for all three groups.

| No. of Sets | Distance (m) | No. of Repetitions | Rest between Repetitions (s) | Rest between Sets (min) |
|:---:|:---:|:---:|:---:|:---:|
| 1 | 10 | 3 | 60 | 4 |
| 2 | 20 | 3 | 90 | 5 |
| 3 | 30 | 3 | 120 | -- |

Additionally, to achieve further physiological input in light of the training, data were collected from five randomly selected players from each of the three groups regarding running speed, heart rate response, and rate of perceived exertion (RPE). In addition to the twice-weekly sprint training sessions, the participants also underwent four specific basketball training sessions per week (two following the sprint training sessions and two on other days of the week). These sessions included shooting drills and technical skill improvements—with and without a ball. These low intensity sessions did not include any

physical fitness exercises. Throughout the research period, the participants were asked to refrain from participating in any other physical training outside the training program.

*Stage 3. Post-intervention tests.* Following the five-week intervention, the tests were repeated to enable assessment and comparison of changes in the players' performance following the intervention.

### 2.2. Tests and Tools

To assess the anaerobic performance of the participants, four tests were conducted throughout the study, including the vertical jump test, the horizontal jump test, the 5 m sprint test, and the T-test. For each test, the participants had two attempts, with 1–3 min rests between them (depending on the test); the higher result of the two was recorded:

The aim of the *vertical jump test* (i.e., Squat Jump [SJ]) was to evaluate *explosive strength in vertical movements*. The participants began by maintaining a half-squat position for 3 s, and then performed a powerful upwards jump—while continuously keeping their hands on their waist [1,2,42].

The aim of the *horizontal jump test* was to evaluate *explosive strength in horizontal movements*. The participants started in a slight squat position with parallel feet, then bent their knees to a half-squat position and pushed themselves forcefully off the ground while extending their arms backward. They then pushed hard against the ground while raising their feet upwards and forwards [1,35]. The achieved distance was measured using a standard measuring tape.

The aim of the *5 m sprint test* was to evaluate explosive power. The participants began from a high position and sprinted as fast as possible in a straight line [1,2].

Finally, the aim of the *t-test* was to examine *agility*. The participants were asked to run as fast as possible between four cones that were placed on the floor in a T-shape. The first cone was placed 10 m away from the starting position, and the additional three cones were placed 5 m away from one another. The participants ran from the first cone to the middle cone, then on to the left cone, then to the right cone, and finally, a *backwards* run to the middle cone. This test entailed movement along multiple planes while facing forwards at all times [42–44].

Speed and agility were measured using a *Witty Wireless Training Timer* (Microgate, Bolzano, Italy), and an *Optojump Next* was used to measure vertical jump performance (Microgate, Bolzano, Italy). The sleds that were used in the intervention groups weighed 28 k and measured 74 × 50 × 80 cm (Bash-Gal, Bnei Brak, Israel).

### 2.3. Statistical Analysis

In this study, the dependent variables included the four anaerobic fitness tests' results. The independent variables included performance in repeated sprints with sled pushing, sled pulling, and no sled. The sample size was calculated using Power Analysis (G*Power version 3.1.9.7) for the explosive power variable (i.e., vertical jump), resulting in the need for a sample size of approximately 30 participants.

Data were analyzed using SPSS v.26 (IBM, Inc., Armonk, NY, USA). Data processing included descriptive statistics and a one-way analysis of variance (ANOVA). First, a one-way ANOVA was conducted to compare the participants' anthropometric data and mean test scores of the tests prior to the intervention to ensure similar starting abilities between groups. Next, the effect of the training program on each group was separately examined using a T-paired sample test for dependent samples. In the third and final stage, the percentage of change between the pre- and post-training periods was compared across the three training groups. For this comparison, the percentage difference in relation to the initial value was calculated: ([post − pre]/pre) × 100. The comparison of the percentage changes among the groups for each dependent variable was conducted using a one-way ANOVA with Fisher's Least Significant Difference (LSD) post hoc analysis.

### 3. Results

As seen in Table 2, no significant differences were found between the groups in any of the anthropometric variables prior to the intervention.

**Table 2.** Anthropometric variables by group (M ± SD).

| Groups | Participants | Control | Pull | Push | *p* |
|---|---|---|---|---|---|
| **Variable** | (*n* = 25) | (*n* = 8) | (*n* = 9) | (*n* = 8) | |
| Age (years ± sd) | 16.7 ± 0.6 | 16.6 ± 0.5 | 16.7 ± 0.8 | 16.8 ± 0.5 | 0.87 |
| Weight (kg ± sd) | 73.6 ± 5 | 73.3 ± 5 | 74.3 ± 6 | 73.1 ± 5 | 0.88 |
| Height (cm ± sd) | 182.6 ± 6 | 182.6 ± 7 | 182.7 ± 7 | 183.1 ± 3 | 0.94 |
| BMI (index ± sd) | 22± 0.9 | 22.1± 0.9 | 22.3 ± 0.6 | 21.8 ± 0.6 | 0.57 |
| Experience (years ± sd) | 9.2 ± 2.1 | 8.5 ± 1.9 | 9.8 ± 2.02 | 9.8 ± 2.4 | 0.32 |

When comparing the players' performance by group prior to the intervention (Table 1), no significant differences were found in any of the variables. Additionally, the data distribution for all variables was normal, except for the speed component. Therefore, for the speed test, a Wilcoxon Signed-Rank Test was performed.

Table 3 presents data regarding the workload and physiological responses during one training session, collected from five participants in each group. Significant differences were seen between groups in all variables (maximum heart rate, running speed at all distances, and RPE), except for average heart rate—where no significant differences were seen.

**Table 3.** Physiological, mechanical, and subjective data by group (M ± SD).

| Variable | HR | Max HR | RPE | Time | | |
|---|---|---|---|---|---|---|
| **Group** | | | (1–10) | 10 m | 20 m | 30 m |
| Control | 107 ± 4.7 | 149 ± 4.3 [†,‡] | 4.5± 0.5 [†,‡] | 1.76 ± 0.07 [†,‡] | 3.26 ± 0.16 [†,‡] | 4.71 ± 0.14 [†,‡] |
| Pull | 115 ± 0.5 | 152 ± 3.9 [*,‡] | 5.3 ± 0.6 [*,‡] | 2.65 ± 0.27 [*,‡] | 4.25 ± 0.11 [*,‡] | 6.16 ± 0.42 [*,‡] |
| Push | 116 ± 4 | 160 ± 4.7 [*,†] | 6 ± 0.2 [*,†] | 3.09 ± 3.15 [*,†] | 5.60 ± 0.11 [*,†] | 7.79 ± 0.27 [*,†] |

*p* < 0.05. [*] Significant difference between the push and pull groups. [†] Significant difference between the push and control groups. [‡] Significant difference between the pull and control groups.

When comparing between the pre- and post-intervention results, a significant improvement (*p* < 0.05) was only seen in the horizontal jump test and in the sled-pushing group (Table 4); a near-significant improvement (*p* = 0.057) was seen in the horizontal jump test in the sled-pulling group. No significant improvements were found for any other variables.

In the 5 m sprint test, no significant changes were found in the two sled groups, yet a non-significant decrease was seen in the running speed in the non-sled group.

Figure 1 provides a graphical representation of the percentage of improvement in each anaerobic performance measure, prior to and following the intervention and by group. No significant difference was found in the percentage improvement between the three groups in any of the performance measures. However, across all examined performance measures, improvements in the two sled groups were consistently higher (though not significant) than in the non-sled control group.

**Table 4.** Comparison of results before and after the intervention by group (M ± SD).

| Group | Control n = 8 | | Pull n = 9 | | Push n = 8 | |
|---|---|---|---|---|---|---|
| Performance measure | Pre | Post | Pre | Post | Pre | Post |
| Sprint 5 m (s) | 1.09 ± 0.1 | 1.14 ± 0.09 | 1.05 ± 0.09 | 1.05 ± 0.11 | 1.04 ± 0.04 | 1.05 ± 0.04 |
| Sprint 20 m (s) | 3.15 ± 0.19 | 3.18 ± 0.13 | 3.06 ± 0.12 | 3.04 ± 0.19 | 3.06 ± 0.08 | 3.03 ± 0.07 |
| Horizontal jump (meter) | 2.24 ± 0.16 | 2.33 ± 0.12 | 2.45 ± 0.17 | 2.52 ± 0.19 | 2.46 ± 0.19 | 2.54 ± 0.21 * |
| Vertical jump (cm) | 35.3 ± 4.22 | 35.5 ± 3.19 | 39 ± 6.47 | 40.93 ± 6.18 | 36.91 ± 5.51 | 37.58 ± 4.31 |
| T-test (s) COD | 10.27 ± 0.2 | 10.16 ± 0.19 | 9.92 ± 0.39 | 9.74 ± 0.51 | 9.92 ± 0.42 | 9.91 ± 0.25 |

* $p < 0.05$.

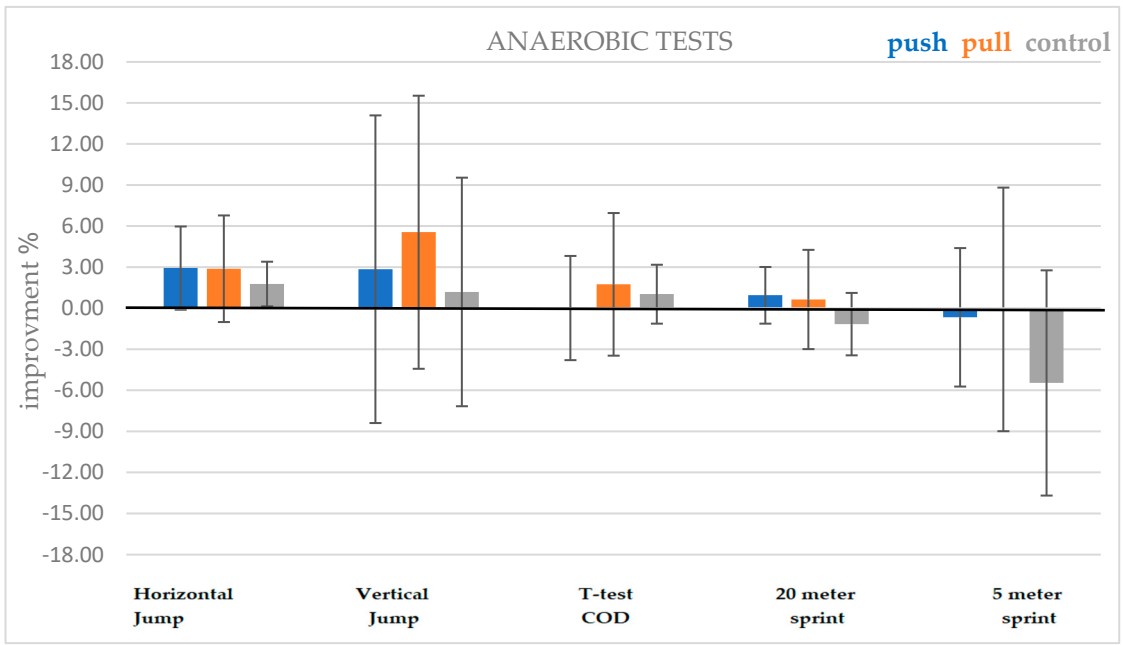

**Figure 1.** Improvements (%) in all tests by group (M ± SD).

## 4. Discussion

The aim of this study was to examine changes in the anaerobic measures of young male basketball players (aged 16–18) following a unique sprint-training intervention using sled pushing, sled pulling, or no sled (body weight only). The anaerobic performance measures that were assessed prior to and following the intervention program included vertical and horizontal jumps, agility, and speed (5 and 20 m sprints). Improvements were seen in the horizontal jump among the sled-pushing team, yet not as hypothesized; no other significant improvements were seen in any other measures or group. However, the sled-training groups exhibited improvements in all measures.

It is important to note that the degree of resistance provided by sleds can be determined in various ways, with the most common method being a percentage of the athlete's body weight [41,45]. Additionally, sled training can be performed with a range of loads, providing light, moderate, or heavy resistance. Given these factors, the inconsistency be-

tween studies is not surprising. In a study on physical education students, improvements in short-distance running speeds were observed after sled-pulling training with a load equivalent to 20% of the athlete's body weight, compared to a load equivalent to 5% or 12.5% in a similar sample that showed significant improvements in vertical jump heights following sled-pulling training, with a moderate 40% load compared to regular sprint training without additional resistance [45].

In the current study, the participants were young male athletes, aged 16–18. Unlike adult populations, adolescents exhibit differences in skeletal development, sexual maturity, muscle mass, and overall body weight [1,15]. As such, sled training loads that are based on the athlete's body weight could lead to significant relative-load discrepancies between the study's diverse participants. For instance, a 40% body weight may constitute a moderate resistance load for one athlete, yet a relatively light load for another younger athlete—potentially leading to varied responses to the sled training [36].

Indeed, Rumpf and Cronin [46] found that sled training with a specific load among athletes with an average age of about 15 years actually slowed down the sprint speed of those who had not yet reached sexual maturity, compared to those who had fully matured [46]. In this study, in an attempt to minimize differences in relative loads between participants, relative weights of only 40% of the participants' weights were used.

Additionally, in this study, significant improvements in the horizontal jump test were only observed in the sled-pushing group, and significant but unclear improvements were only found in the vertical jump test in both sled groups. No significant improvements were found in either the sprint tests or post-intervention agility tests, in any group. The lack of significant improvements in most performance measures in the current study, compared to positive findings in previous studies, could stem from various reasons. First, the training period implemented in the current study was relatively short (about one month), compared to longer intervention periods seen in other studies (two–three months) [36,40,47]. The five-week intervention applied in this study may not have been sufficient for inducing significant changes in the young participants; the small team size and inclusion–exclusion criteria limited the number of eligible participants. Further studies are needed to improve the generalizability of the results, including a larger sample size and a control group [48].

Moreover, sled pushing and pulling resistance exercises provide a less conventional training modality than the bodyweight jumps and short, fast sprints to which the participants in this study were accustomed. With sleds, athletes need to develop a specific technique for running against such resistance, as well as exerting power for overcoming the additional weight and initiating movement from a static position. It is possible that an adaptation period—with an emphasis on technique learning, based on running with added yet light resistance (10–20% body weight) prior to the intervention—may have been beneficial.

Sled training and added weight could teach athletes how to maintain the desired body position while pushing as a means for increasing horizontal force. Yet athletes who are new to sled pushing tend to lean overly forward—potentially negatively affecting their running technique during acceleration. Moreover, as the athletes' hands must grip the push handles, they cannot use these limbs to enhance body balance during the sprint [40,49]. Interestingly, in a study on professional rugby players, the findings revealed that the greater the sled weight, the smaller the stride lengths and the greater the forward lean—leading to higher horizontal forces [41].

Although non-significant, improvements were seen in the current study in all measures in both sled-training groups compared to those in the control group. This adds to the literature whereby sled training contributes to anaerobic performance. In the game of basketball, which primarily relies on anaerobic performance, players could especially benefit from sled training. Movements specific to the game, such as rebounds, agility dribbles, blocks, changes of direction, and fast breakouts, could enhance players' productivity and contribution to the team, following the use of this specific training tool.

A more detailed examination of the impact of sled training, especially the comparison between the various performance measures tested in the current study, indicated that improvements in the horizontal and vertical jump tests were greater than improvements in the sprint and agility tests. This suggested that sled training may be more effective for improving explosive strength—characterized by cyclical movements such as jumps, compared to improving speed—characterized by cyclical movements such as running. A study supporting this assumption found that the heavier the sled, the lower the step frequency, yet the higher the pushing force [41]. Sprinting speed is derived from both step frequency and step length, while jump height is mainly derived from pushing force; as such, the larger change observed in the jump tests in this study compared to that in the sprint tests appears reasonable. However, this assumption was not conclusively supported in the current study.

*Limitations and Future Research*

In addition to a relatively short intervention period (five weeks), the small number of participants in each group (8–9) in the current study may have hindered the demonstration of positive changes in the tested variables. It should be noted that additional athletes were included at the onset of the study, yet this number gradually declined over the weeks, mainly due to injury, illness, or dropout. Future studies could benefit from including a larger number of participants. Moreover, in addition to conducting a sled-training intervention for a longer period, an adaptation period prior to the intervention should also be introduced, during which time the participants become familiar with sled-related techniques. Future studies should also assess the participants' sexual maturity, through Tanner stage assessments, to achieve additional insights into sled training in athletes in general and in basketball players in particular.

## 5. Conclusions

This study found that employing sleds, either pulling or pushing, with a resistance equivalent to 40% of the athlete's body weight did not lead to significant improvements in most anaerobic performance measures. Only sled-*pushing* training showed notable enhancements in horizontal jumps compared to sled-pulling and non-sled training. Limited changes across the three groups could be attributed to the relatively short intervention period and the relatively small number of participants.

Despite limited observable changes, the groups that trained with sleds exhibited more consistent improvements in all measured variables compared to the non-sled training group. Both sled training groups consistently demonstrated higher improvements in cyclic movements, such as vertical and horizontal movements, crucial for power development in activities such as jumps, blocks, directional changes, stops, and leaps. This emphasizes the potential benefit of sled resistance as a tool for enhancing anaerobic performance in young male athletes, including basketball players.

However, sled training should be part of a comprehensive training approach, tailored to the specific needs of the sport, athletes, roles, and training periods. When working with young, pre-sexually mature athletes, caution should be taken. Indeed, sled usage should be gradual and mindful of their age, weight, and developmental stage.

**Author Contributions:** Conceptualization, R.G. had the original idea of the paper and wrote the paper; investigation, A.L. and A.S.; supervision, Y.M. and J.C.G. were the directors and gave the final approvals of the text. All authors have read and agreed to the published version of the manuscript.

**Funding:** This research received no external funding.

**Informed Consent Statement:** Informed consent was obtained from all subjects involved in the study; for participants under the age of 18, informed consent was also obtained from one of their parents. They were also free to withdraw from the study at any time, without having to provide an explanation for this decision. Although anonymity could not be ensured due to the nature of the study, the participants and their parents were assured that the utmost confidentiality and scientific rigor would be applied throughout the study and that the acquired data would only be used for the purpose of this research project.

**Data Availability Statement:** The data presented in this study are available on request from the corresponding author and the first author. The data are not publicly available due to ethical and privacy restrictions.

**Acknowledgments:** The authors would like to thank the members of the Male Youth Department at the Maccabi Rishon Lezion Basketball Club, including the managers, coaches, trainers, and players for their participation in this study.

**Conflicts of Interest:** The authors declare no conflicts of interest.

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
