# Peer review of "The Use of Sleds as a Unique Training Technique for Anaerobic Performance Development among Young Basketball Players"

_applsci, doi:10.3390/app14072696_

Round 1

Reviewer 1 Report

Comments and Suggestions for Authors

(Overall) I think it is a meaningful study to understand the effectiveness of the unique training technique for young basketball players' anaerobic performance development. I think it will be possible to develop better research results if the reviewer makes small revisions as suggested below.

(Introduction) Overall, the introduction is well organized. However, the suggestion of the necessity of this study based on previous studies is somewhat insufficient. After reorganizing the preceding studies and deriving the gap, please clearly present the reason why this study is necessary.

(Hypothesis) Researchers have explained that there are two hypotheses, but the actual hypotheses are presented in three ways. Please correct them.

In the (Materials and Methods) research method, please present the process of conducting the experiment in this study as one <Figure> so that it can be understood at a glance.

(Table) The <table> form is different from that of the journal. Please check and present the editorial instructions of the journal.

(Reference) The format in which the reference is presented does not match the format in the journal. Please check and present the editorial instructions in the journal.

(Reference) There are many references in which the name of the journal is abbreviated (ex, Eur J Sport Sci). Please find them all (European Journal of Sport Science).

Author Response

Reviewer response 1

(Overall) I think it is a meaningful study to understand the effectiveness of the unique training technique for young basketball players' anaerobic performance development. I think it will be possible to develop better research results if the reviewer makes small revisions as suggested below.

Thank you for the positive feedback!

(Introduction) Overall, the introduction is well organized. However, the suggestion of the necessity of this study based on previous studies is somewhat insufficient. After reorganizing the preceding studies and deriving the gap, please clearly present the reason why this study is necessary.

The article is preliminary in the population of young basketball players with new technology and original methodology.

(Hypothesis) Researchers have explained that there are two hypotheses, but the actual hypotheses are presented in three ways. Please correct them.

Thank you for the comment, I correct it.

The following two research hypotheses were defined:

  1. Significant improvements will be seen in the participants’ anaerobic abilities following the sled training, similar improvements will be seen in the anaerobic capacity of the two intervention groups compared to the control group.
  2. The impact of sled pushing and pulling at 40% of body mass will have a higher effect compared to repetitions without sled use on anaerobic components such as explosive power, maximum speed, and agility among young basketball players. There won't be a difference in the impact between sled pushing and sled pulling workouts among the players.

In the (Materials and Methods) research method, please present the process of conducting the experiment in this study as one <Figure> so that it can be understood at a glance.

Thank you very much! We presented it in the text, therefore we thought it unnecessary to add the information in a figure.

(Table) The <table> form is different from that of the journal. Please check and present the editorial instructions of the journal.

Thank you, I correct it

(Reference) The format in which the reference is presented does not match the format in the journal. Please check and present the editorial instructions in the journal.

(Reference) There are many references in which the name of the journal is abbreviated (ex, Eur J Sport Sci). Please find them all (European Journal of Sport Science).

Thank you, but I recently published an article in MDPI and the bibliography is listed with abbreviations according to the requirements of the journal.

Reviewer 2 Report

Comments and Suggestions for Authors

A well-made article, with pertinent bibliographic sources, which follows a coherent logical thread. An article that, through its conclusions, saves the time of young trainers by limiting the waste of time for (apparently favorable) means that do not bring great benefits in the training process.

Author Response

Reviewer response 2

A well-made article, with pertinent bibliographic sources, which follows a coherent logical thread. An article that, through its conclusions, saves the time of young trainers by limiting the waste of time for (apparently favorable) means that do not bring great benefits in the training process.

Thank you for the positive feedback!

Reviewer 3 Report

Comments and Suggestions for Authors

The introductory section is really long and distracting, you should provide some background, but a slim rationale for the L188-203 aim. among other things, the objective seems to be the basis for a scoping review of the literature...

L 206 I would omit the inclusion of 25 athletes. It is a result and must be inserted in the appropriate section, describing the selection pathway.

In fact, eligibility should be made clearer to the reader. Is Rishon LeZion a division? what level? did you only include one team? what types of injuries? in-season? off-season?

L221 could you provide some figures about it?

L307 by convention you should provide the reader with the selection path of the sample included in the study

L311 the acronyms are not explained

L330 the tables are literally formatted images.. the references to each metrature (10,20,30) are not clear..

For fig 1, I recommend providing appropriate box and whisker plots...

In discussion you might provide: "A global rehabilitative approach based on education to avoid upper limb pain injuries, training, and improvement of ergonomics through kinematic analysis, was shown to play a key role in improving upper limb function in male basketball players" ref: Demeco A, de Sire A, Marotta N, Palumbo A, Fragomeni G, Gramigna V, Pellegrino R, Moggio L, Petraroli A, Iona T, et al. Effectiveness of Rehabilitation through Kinematic Analysis of Upper Limb Functioning in Wheelchair Basketball Athletes: A Pilot Study. Applied Sciences. 2022; 12(6):2929. https://doi.org/10.3390/app12062929

Author Response

Reviewer response 3

The introductory section is really long and distracting, you should provide some background, but a slim rationale for the L188-203 aim. among other things, the objective seems to be the basis for a scoping review of the literature...

Thank you for the positive feedback and for the minor corrections ! 

L 206 I would omit the inclusion of 25 athletes. It is a result and must be inserted in the appropriate section, describing the selection pathway.

Thank you for the comment, but it should present in the methods part 

In fact, eligibility should be made clearer to the reader. Is Rishon LeZion a division? what level? did you only include one team? what types of injuries? in-season? off-season?

 A total of 25 male basketball players aged 16–18yrs participated in the study They were members of the  2 youth teams of the Rishon LeZion Basketball Association in Israel and had 8–9yrs of basketball experience. For the past four years, in addition to their basketball training, they had also participated in two weekly 60min strength-training sessions with weights in the gym, and in one weekly 45min athletic training sessions. All players followed the same training regime, and none had no previous experience in sled training. The inclusion criteria were as follows: (1) no physical injury over the two months leading up to or during the intervention; (2) had participated in at least 80% of the training sessions in the two months leading up to or during the intervention; and (3) had at least 5yrs of basketball experience.

Stage 1. Pre-intervention tests. On the first week of the preseason, the participants underwent anaerobic fitness tests in the facility where they regularly trained.

L221 could you provide some figures about it?

Thank you, do you mean for the tests? Do you think it is necessary?

L307 by convention you should provide the reader with the selection path of the sample included in the study

Thank you,

L311 the acronyms are not explained

Thank you, but I don't see any  acronyms in this line

L330 the tables are literally formatted images.. the references to each metrature (10,20,30) are not clear..

Thank you, I correct it   

For fig 1, I recommend providing appropriate box and whisker plots...

Thank you  

In discussion you might provide: "A global rehabilitative approach based on education to avoid upper limb pain injuries, training, and improvement of ergonomics through kinematic analysis, was shown to play a key role in improving upper limb function in male basketball players" ref: Demeco A, de Sire A, Marotta N, Palumbo A, Fragomeni G, Gramigna V, Pellegrino R, Moggio L, Petraroli A, Iona T, et al. Effectiveness of Rehabilitation through Kinematic Analysis of Upper Limb Functioning in Wheelchair Basketball Athletes: A Pilot Study. Applied Sciences. 2022; 12(6):2929. https://doi.org/10.3390/app12062929

Thank you for the reference, I add it to the discussion part